# Mitigating Vascular Inflammation by Mimicking AIBP Mechanisms: A New Therapeutic End for Atherosclerotic Cardiovascular Disease

**DOI:** 10.3390/ijms251910314

**Published:** 2024-09-25

**Authors:** Jun-Dae Kim, Abhishek Jain, Longhou Fang

**Affiliations:** 1Center for Cardiovascular Regeneration, Department of Cardiovascular Sciences, Houston Methodist Research Institute, Houston, TX 77030, USA; 2Department of Biomedical Engineering, College of Engineering, Texas A&M University, College Station, TX 77843, USA; a.jain@tamu.edu; 3Weill Cornell Medical College, Cornell University, Ithaca, NY 14850, USA

**Keywords:** ASCVD, vascular inflammation, inflamed EC-specific targeting, AIBP

## Abstract

Atherosclerosis, characterized by the accumulation of lipoproteins and lipids within the vascular wall, underlies a heart attack, stroke, and peripheral artery disease. Endothelial inflammation is the primary component driving atherosclerosis, promoting leukocyte adhesion molecule expression (e.g., E-selectin), inducing chemokine secretion, reducing the production of nitric oxide (NO), and enhancing the thrombogenic potential. While current therapies, such as statins, colchicine, anti-IL1β, and sodium–glucose cotransporter 2 (SGLT2) inhibitors, target systemic inflammation, none of them addresses endothelial cell (EC) inflammation, a critical contributor to disease progression. Targeting endothelial inflammation is clinically significant because it can mitigate the root cause of atherosclerosis, potentially preventing disease progression, while reducing the side effects associated with broader anti-inflammatory treatments. Recent studies highlight the potential of the APOA1 binding protein (AIBP) to reduce systemic inflammation in mice. Furthermore, its mechanism of action also guides the design of a potential targeted therapy against a particular inflammatory signaling pathway. This review discusses the unique advantages of repressing vascular inflammation or enhancing vascular quiescence and the associated benefits of reducing thrombosis. This approach offers a promising avenue for more effective and targeted interventions to improve patient outcomes.

## 1. Introduction

Increased systemic inflammation significantly elevates the risk of atherosclerotic cardiovascular disease (ASCVD). While therapies that lower inflammatory cytokines can reduce this risk, the systemic suppression of inflammation can lead to adverse side effects, such as dysregulated immune responses to infections, which can have lethal consequences. Therefore, an optimal therapy would be specific to a particular tissue and would minimize side effects.

A critical factor exacerbating atherosclerosis is vascular inflammation and the associated detrimental effects [1,2,3,4]. AIBP has emerged as a critical regulator of angiogenesis and inflammation. Beyond its well-documented extracellular roles, recent findings highlight its intracellular functions, particularly in mitochondrial regulation. This review will examine potential therapeutic strategies for leveraging existing pharmaceuticals and AIBP to mitigate endothelial inflammation and will propose that targeted therapies directed at the inflamed endothelium could further diminish the risk of ASCVD in the post-statin era.

## 2. The Initial Cause of Vascular Inflammation

The initial trigger of vascular inflammation is often instigated by lipoprotein retention in the subendothelial space and subsequent modifications that render them proinflammatory. This process relies on several key mechanisms (Figure 1A).

### 2.1. Disrupted Vascular Integrity

In regions where endothelial junctions are compromised, lipoproteins may enter the subendothelial spaces. Although the size of LDL particles is large (~500 kDa) and, thus, this route may represent a minor pathway, it may provide an important channel of entrance for smaller size high-density lipoproteins, or lipid-poor APOA1. In advanced atherosclerotic lesions, vascular erosion and the resultant loss of endothelial junctions can exacerbate inflammation.

### 2.2. Enhanced Lipoprotein Transcytosis

Transcytosis is a key pathway for LDL penetration into the subendothelial space. The putative HDL receptor that is abundantly expressed in the liver, Scavenger Receptor Class B Type I (SR-BI), and other such receptors play an unexpected role in endothelial SR-BI transcytosis. The loss of endothelial SR-BI is, thus, atheroprotective in hyperlipidemic mice. SR-BI regulates the guanine nucleotide exchange factor dedicator of cytokinesis 4 (DOCK4) [5], which promotes SR-BI internalization of and LDL transport by activating (Ras-related C3 botulinum toxin substrate 1) RAC1. RAC1 is a small GTPase of the Rho family, involved in controlling the organization of the actin cytoskeleton. In addition, (activin receptor-like kinase 1) ALK1 was identified in an unbiased screening as another important receptor for LDL transcytosis [6], mediating LDL uptake into endothelial cells (ECs) via a non-degradative pathway. ALK1 is a part of the transforming growth factor-beta (TGF-β) receptor family and is predominantly expressed in ECs, where it plays a crucial role in regulating vascular development. Compared with LDLR, ALK1 binds LDL with lower affinity. HDL can similarly undergo transcytosis into the aortic endothelium, mediated by F0F1 ATPase/ATP synthase binding to APOA1 [7,8]. However, the physiological significance of F0F1 ATPase-mediated APOA1/HDL uptake is unclear. It will be interesting to determine whether an aortic extracellular F0F1 ATPase blockage can analogously reduce the atherosclerotic burden, as an endothelial SR-BI deficiency. Indirect evidence suggests possible HDL sub-endothelial retention, as oxidized APOA1 is also found inside atherosclerotic lesions [9,10]. Utilizing fluorescent HDL to monitor its tissue distribution would provide direct evidence.

### 2.3. Disturbed Shear Stress

Low fluid shear stress, resulting of oscillatory flow at the bifurcation of the artery, is well-known to elicit inflammation. Recent studies show that disturbed flow can elicit vascular inflammation, even without hyperlipidemia. These vascular locations of low shear stress correspond to the later accumulation of atherosclerotic lesions. While many receptors are proposed to mediate the force of laminar flow, the sensor responsible for low shear stress remains less well-known. The (Yes-associated protein) YAP/(transcriptional coactivator with PDZ-binding motif) TAZ has been proposed as the transcriptional responder to low flow, activating a proinflammatory program [11]. For instance, YAP binds (NOD-like receptor family pyrin domain-containing 3) NLRP3 [12], the inflammasome, and increases NLRP3 stability by blocking its proteasomal degradation. However, in the context of inflammatory bowel disease, instead YAP inhibits the inflammatory response in colitis. YAP can physically interact with (Enhancer of Zeste Homolog 2) EZH2, which sustains H3K27me3 enrichment on the promoter of (Jumonji domain-containing protein 3) JMJD3 [13], a histone H3K27me3 demethylase. As a result, it reduces JMJD3 expression and attenuates proinflammatory JMJD3 signaling.

### 2.4. Cholesterol Crystals

The plasma membrane of mammalian cells contains abundant cholesterol levels, accounting for 10–30% of total lipids [14]. Under hypercholesterolemia, cholesterol crystals can form at the cell surface of ECs, VSMC and macrophage foam cells [15,16,17], eventually contributing to plaque instability [18]. Interestingly, caveolin 1 levels are increased under hypercholesterolemia, further elevating cell surface cholesterol and inhibiting endothelial nitric oxide synthase (eNOS) [19]. In addition to its putative role in eNOS regulation, CAV1 also mediates LDL transcytosis. CAV1 knockout reduces LDL transcytosis of the endothelium and ameliorates endothelial inflammation [20]. Using a hyperlipidemic rabbit model, cholesterol content is found to be progressively enriched on the plasma membrane of aortic ECs [21]. When cholesterol levels in the plasma membrane reach a critical threshold, typically around 30–50 mole percent (mol%), it can exceed the solubility limit of cholesterol in the phospholipid bilayer. This can lead to lipid phase separation, where cholesterol-rich domains (often referred to as lipid rafts) form within the membrane. If cholesterol saturation continues to rise beyond this point, it could potentially result in the formation of cholesterol crystals or patches, which may be irreversible under certain conditions [22,23]. These initial immiscible cholesterol domains on the cell surface can build up more cholesterol crystals [24], the formation of which are dictated by the content of free cholesterol. Enhancing cholesterol esterification or free cholesterol efflux can reduce crystal generation. In addition, oxidative stress can also enhance cholesterol crystal formation. Cholesterol crystals can be present in distinct shapes (e.g., needle, plate, helix) under hyperlipidemic conditions [16,23]. They can form in distinct subcellular locations, such as intracellularly, following lipoprotein particle uptake and final lysosomal accumulation [15]. Enhanced autophagy and cholesterol esterification are suggested as the key pathways to reduce cholesterol crystal formation in aortic ECs.

The zebrafish model, due to optical transparency in the early stage, has been used to study vascular inflammation. Zebrafish, without any genetic modification, develop hyperlipidemia following approximately two weeks of high-fat diet feeding, leading to greater leukocyte recruitment to the vasculature [25,26]. Supplementation with the fluorescent Bodipy lipid allows the observation of lipid droplets on the vascular wall. In contrast to mammals, in zebrafish larvae most lipid droplets are found in the cardinal vein [27]. One possibility is that this location in the zebrafish larva stage functions as a primitive immune organ or has an unusual phagocytic capacity. Transgenic zebrafish overexpressing the single-chain antibody (IK17) against oxidized LDL epitopes can bind these venous vascular lesions [28]. Whether the venous lipid accumulation is the zebrafish equivalent of mammal fatty streaks is a question that needs further investigation.

## 3. Therapies Reducing Systemic Inflammation

### 3.1. Lipid-Lowering Therapeutics

Statins are identified as inhibitors of the rate-limiting enzyme for cholesterol biosynthesis, 3-hydroxy-3-methylglutaryl coenzyme A (HMG-CoA) reductase. Statins reduce plasma LDL cholesterol (LDL-C) by increasing the expression of LDLR, a compensatory activation of sterol regulatory element binding protein 2 (SREBP2), the master transcription factor for cholesterol biosynthesis. While statins reduce mortality associated with high-risk patients with cardiovascular disease [29], they also exert a substantial anti-inflammatory function [30]. A meta-analysis of 15 randomized clinical trials revealed that statin treatment lowered the content of endothelin-1, a potent vasoconstrictor peptide produced by ECs in response to inflammation. Statins also improve vascular function by reducing the plasma content of asymmetric dimethylarginine (ADMA) [31], an inhibitor of NO synthase, and lowering CAV-1 levels, which in turn de-represses eNOS inhibition. Treatment of hyperlipidemic animals with atorvastatin lowers systemic levels of proinflammatory cytokines, while increasing the abundance of anti-inflammatory IL-10 [32]. In the JUPITER randomized clinical trial, approximately 17,000 otherwise healthy subjects with LDL-C >130 mg/dL and CRP >2 mg/L were treated with either 20 mg daily of rosuvastatin or a placebo. This treatment reduced LDL-C by 50% and the CRP level by 37%, leading to reduced risk of MI, stroke, and all-cause mortality [33]. Two meta-analyses of human clinical trials have validated that statin treatment in combination with ezetimibe, the pharmacological inhibitor of intestinal cholesterol reabsorption, reduces CRP, IL6, and TNFα [34,35].

Two mechanisms contribute to the anti-inflammatory effects of statins, as follows:i.Effects on Inflammatory Signaling. Lipid rafts are often the platform that facilitates inflammatory signaling. Statins reduce the lipid raft content in ECs. Recently, Miller and colleagues proposed the concept of inflammarafts [36], where major proinflammatory receptors cluster in lipid rafts and sustain inflammatory signaling. Statin-mediated inhibition of cholesterol synthesis is predicted to reduce the signaling potency of inflammarafts [37]. Statin treatment also impacts the synthesis of isoprenoids, critical intermediates that facilitate the post-translational modification and functional activation of various proteins involved in inflammatory responses, such as small GTPases (Rho, Rac, and Ras). By restricting the prenylation of these molecules, statins impair the signaling pathways that lead to the activation of NF-κB, a key transcription factor in the expression of proinflammatory cytokines [38];ii.Reduction in Cytokine Production. Statins lower the levels of proinflammatory cytokines, such as IL-6, TNF-α, and IL-1β. This involves both the inhibition of cytokine synthesis in immune cells and the reduction of their expression, through the modulation of transcription factors like NF-κB. This decrease in cytokine production mitigates inflammatory responses, particularly in vascular tissues;iii.Enhancement of the Endothelial Function. The endothelium plays a central role in maintaining vascular homeostasis, and its dysfunction precedes and contributes to atherosclerosis and other cardiovascular diseases. Statins promote endothelial health by increasing the bioavailability of NO, a vasodilator with anti-inflammatory properties, through the upregulation of eNOS activity [39]. Improved endothelial function contributes to reduced vascular cell adhesion molecule (VCAM) expression and leukocyte adhesion, decreasing vascular inflammation. The reversal of endothelial dysfunction in human vessels by statins is dependent on the mevalonate pathway and Rac1 inhibition. These critical steps reduce NADPH oxidase activity and improve tetrahydrobiopterin bioavailability and nitric oxide synthase (NOS) coupling in human vessels [40];iv.Plaque Stabilization and Immune Modulation. Statins contribute to the stabilization of atherosclerotic plaques by reducing the infiltration of macrophages and other inflammatory cells, thus lowering the risk of plaque rupture [41]. Additionally, statins influence T cell differentiation and function, promoting the shift from a proinflammatory Th1 phenotype to an anti-inflammatory Th2 phenotype [42]. This modulation of the immune response is crucial in dampening systemic inflammatory responses [43].

In addition to statins, other lipid lowering therapies also reduce systemic inflammation. Proprotein convertase subtilisin/kexin 9 (PCSK9) is a chaperone that targets LDLR for lysosomal degradation. Both genetic and antibody-mediated neutralization convincingly show a deficiency in PCSK9 lowers plasma LDL-C [44,45,46]. Interestingly, PCSK9 functions beyond cholesterol regulation; it also precipitates inflammation [47]. Molecularly, PCSK9 regulates the TLR4/NFkB signaling pathway. The PCSK9 gain-of-function mutation is positively associated with the upregulation of TLR4, a key type I membrane receptor controlling innate immune responses [48]. This results in the activation of its downstream target NF-kB, an effect that can be rescued by PCSK9 ablation. PCSK9 contains a unique C-terminal cysteine-rich domain, which is implicated in binding and activating TLR4 [47].

Ezetimibe, the inhibitor of intestinal Niemann–Pick C1-Like 1 (NPC1L1), which is responsible for cholesterol reabsorption, is also applied to reduce plasma LDL-C levels. While ezetimibe is often used in combination with other lipid-lowering drugs, ezetimibe alone reduces the atherogenic burden. In addition to lowering the plasma total cholesterol level, ezetimibe reduces circulating inflammatory markers. However, ezetimibe does not reduce the content of oxidized LDL, an important proatherogenic component in plasma [49].

### 3.2. Colchicine

Colchicine is an anti-inflammatory medicine used to treat acute gout. Molecularly, colchicine binds to tubulin and inhibits its polymerization, which impedes cytoskeletal rearrangement, resulting in attenuated chemotaxis and cell signaling. In addition, it can reduce the expression of inflammatory adhesion molecules, TNF-α elicited NF-kB signaling, and inflammasome activation [50,51]. Several clinical studies have explored the protective role of colchicine in cardiac function immediately following MI. The randomized COLCOT trial recruited 4745 patients to receive either colchicine or a placebo within 30 days post-MI, and the colchicine group significantly lowered primary endpoint, including MI, stroke, and cardiovascular mortality. However, an adverse effect of pneumonia was found to be increasingly present in the colchicine group [52]. The improvements were also validated by other smaller-scale, randomized clinical studies [53].

Another large-scale randomized clinical study, the LoDoCo-2 trial, involved 5522 patients with chronic coronary artery disease, who were given either colchicine or a placebo. Consistent with the COLCOT trial, the colchicine treatment group showed significantly fewer cases of primary endpoints, such as MI, stroke, and cardiovascular mortality, and coronary revascularization following ischemia. In contrast, the mortality of non-cardiovascular disease was increased in the treatment group [54]. Despite the adverse effect, the beneficial impact of colchicine on reducing the risk of chronic coronary artery diseases has been validated in meta-analysis and clinical trials of a smaller size [55].

Despite the above stated benefits, colchicine treatment also has limitations: it has not shown a significant mortality benefit and some studies indicate an increased risk of non-cardiovascular deaths, particularly from infections [51]. In addition, colchicine’s gastrointestinal side effects and its potential to interfere with other medications pose challenges, especially in older or multimorbid patients [56]. These limitations highlight the need for further research to better define colchicine’s safety profile and optimal use in diverse patient populations.

### 3.3. Anti-IL1β Therapy

IL1β levels are associated with CAD, but the therapeutical significance of targeting IL1β was once doubted. Canakinumab, a human monoclonal antibody IL1β, has been approved for the treatment of various autoimmune diseases [57]. Canakinumab delivery reduces IL-6 and CRP levels in plasma, but has modest effects on the lipid profile. A clinical trial, known as the CANTOS trial, tested the therapeutic effect of IL1β neutralization on CAD. A total of 10,061 patients with CRP levels ≥ 2 mg/L and a previous MI were assigned to receive canakinumab or a placebo. The doses tested were 50, 150, and 300 mg, every three months via a subcutaneous injection. All three doses reduced high-sensitivity CRP levels by 26% to 41%. Canakinumab did not change the lipid profile, but reduced the incidence rate of the primary endpoint. However, only the 150 mg dose, but not the 50 or 300 mg doses, significantly reduced the primary and the secondary endpoint. While no significant difference in all-cause mortality was found, canakinumab treatment was associated with a higher incidence of fatal infection, such as sepsis [57]. This clinical study convincingly documents the causal role of IL1β-mediated inflammatory signaling in non-fatal MI, stroke, and cardiovascular mortality, following an initial MI. The increased risk of fatal infections in a subset of patients suggests that the exclusion of vulnerable sub-populations would be preferred. Furthermore, while Canakinumab does not reduce plasma lipid levels, its impact on atherosclerotic plaque composition and the long-term effects of IL1β inhibition remain unclear. If repression of inflammatory signaling represents a key therapeutic direction, it is worth investigating whether there would be additive benefits from simultaneously inhibiting other inflammatory signaling pathways.

## 4. SGLT2 Inhibitors

The sodium–glucose co-transporter-2 (SGLT2) protein is located in the proximal tubule of human nephrons, where it regulates the reabsorption of glucose from the filtrate back into the bloodstream. The recent development of SGLT2 inhibitors (SGLT2i) has shown exciting therapeutic applications in CVD. Interestingly, SGLT2 inhibition reduces major adverse CV events, including non-fatal MI, non-fatal stroke, and the composite endpoint of CV death, independent of diabetes mellitus. While some of the protective effects of SGLT2i are mediated through their action on renal tubules, others also function independently. For example, SGLT2i empagliflozin documents cardioprotective effects in mice, even in the absence of renal SGLT2 expression [58].

Mechanistically, SGLT2i can confer protective effects on CVD by directly improving the endothelial function, such as mitigating vascular inflammation, reducing oxidative stress, increasing NO bioavailability, and augmenting mitochondrial function. SGLT2i treatment results in lower levels of proinflammatory cytokines, namely CRP, TNFα, IL6, and MCP-1, presumably via inhibiting NF-kB activation [59,60]. In addition, it enhances the profile of vascular quiescence by improving vascular junctions, reducing the expression of VCAM-1, ICAM-1, and E-selectin, as well as dampening NLRP3 inflammasome activation, collectively limiting immune cell recruitment [61]. Another mechanism of action according to which SGLT2i reduces inflammation is by reducing renal urate absorption, thereby lowering proinflammatory uric acid levels in the blood [62]. However, SGLT2i does not show a protective effect against thrombosis, which underlies heart attacks and strokes [63]. SGLT2i can also improve NO levels via increasing AMPK activation, which in turn augments eNOS activity, thereby improving endothelial function [64]. Other studies show that SGLT2i inhibits NADPH oxidase activation, reducing endothelial oxidative stress [65]. Furthermore, SGLT2i can extend the bioavailability of NO by repressing the mechanism that degrades NO [66]. SGLT2i is postulated to preserve mitochondrial genetic integrity by enhancing AMPK activation and reducing DRP activation, leading to physiological mitochondrial fission [67].

Mitophagy, the process of removing damaged mitochondria via autophagy and functioning as mitochondrial quality control, can also be enhanced by SGLT2i, resulting in improved mitochondrial function. Dysregulated mitophagy contributes to atherogenesis, and SGLT2i can improve mitophagy by inhibiting the mTOR and SIRT1 pathways, reducing atherosclerotic burden [68,69]. SGLT2i can improve vascular health by enforcing a metabolic shift that enhances the utilization of fatty substrates, independent of changes in insulin levels [70].

## 5. Eicosanoids

Eicosanoids, particularly leukotrienes produced via the 5-lipoxygenase (5-LO) pathway, such as leukotriene B4 (LTB4) and leukotriene C4 (LTC4), regulate inflammatory responses in atherosclerosis. Modulating the leukotriene pathway has been demonstrated to attenuate atherosclerotic progression in animal models, underscoring their potential as therapeutic targets. Previous research has highlighted the value of cysteinyl leukotrienes (CysLTs) as biomarkers in the context of peripheral arterial disease (PAD) [71]. PAD is caused by atherosclerosis, which restricts blood flow to the lower extremities. Specifically, this study investigated the urinary levels of leukotriene E4 (uLTE4), a stable metabolite of LTE4, in patients with PAD undergoing percutaneous transluminal angioplasty (PTA). The findings revealed that elevated uLTE4 levels, assessed both pre- and post-PTA, are significantly associated with restenosis and re-occlusion [71]. Therefore, uLTE4 appears to be a predictive biomarker for vascular complications following angioplasty.

Elevated levels of inflammatory leukotriene E4 (LTE4), thromboxane B2 (TXB2), but not 5-Hydroxyeicosatetraenoic acid (5-HETE), results in lower quality-of-life scores, suggesting that vascular inflammation influences disease outcomes. These results warrant further exploration into the therapeutic potential of reducing eicosanoid levels to mitigate vascular inflammation and improve clinical outcomes in PAD [72]. Targeting endothelial eicosanoids, however, presents significant challenges because of their widespread distribution across various tissues. Their release into the extracellular space exerts complex and diverse biological effects.

## 6. AIBP

AIBP is a secreted protein that was first implicated in spermatogenesis [26], identified in yeast two-hybrid screen for proteins that bind APOA1. Surprisingly, mice with AIBP deletion are perfectly fertile. The function of AIBP began to emerge when a proteome blast using the MD2 protein, a co-receptor of TLR4, retrieved AIBP. Subsequent studies support the conclusion that AIBP binds TLR4 and functions to regulate TLR4 signaling. Instead of supporting TLR4 signaling, AIBP restricts it. Mechanistically, this involves lipid rafts, the key microdomains that facilitate cell surface receptor signaling (Figure 1B). As the plasma membrane is enriched with cholesterol, these transmembrane receptors tend to cluster in cholesterol-enriched microdomains, propagating their signaling. The advantage of clustering is likely to increase the local concentrations of signaling components for efficient signaling. AIBP promotes cholesterol efflux to APOA1 and HDL. By binding to TLR4, AIBP can create a cholesterol-reduced microdomain in the vicinity of TLR4, which is postulated to disrupt lipid rafts and impede TLR4 signaling. The mechanism by which AIBP promotes cholesterol efflux is still unclear.

Our prior studies suggest that AIBP can increase the maximal binding capacity of HDL to ECs and, at the same time, enhance the turnover rate of HDL (on/off rate from the plasma membrane), thereby increasing EC cholesterol efflux [73,74]. Interestingly, a recent paper indicates that AIBP, binding phosphatidylinositol 3-phosphate (PI(3)P), promotes CDC42-dependent lipid raft disruption [75]. This mechanism requires cytoskeletal rearrangement. AIBP incubation also increases endocytosis (Figure 1C) and its subsequent co-localization with the early endosome [75]. It is tantalizing to speculate that endocytosis contributes to AIBP-mediated lipid raft disruption, given an earlier study reporting the crucial role of endocytosis in mediating cholesterol efflux to APOA1 [76]. The role of this mechanism in cholesterol efflux is further supported by the findings from the Miller laboratory, using an AIBP mutant that does not bind TLR4 [77]. Interestingly, this AIBP mutant retains the capacity to disrupt lipid rafts. However, a kinetic study is warranted to determine whether this AIBP mutant functions as efficiently as wild-type AIBP in promoting cholesterol efflux and disrupting lipid rafts. Another possibility is that there is an additional cell surface binding partner of AIBP, compensating for its binding to TLR4. However, TLR4 deficiency appears to blunt all AIBP binding to macrophages, arguing against the existence of another AIBP binding partner in macrophages. It should be noted that cholesterol efflux, by reducing the free cholesterol content from cells, may be inherently anti-inflammatory. This is because free cholesterol is toxic when exceeding threshold levels of maximal solubility in the plasma membrane [78]. It is not unexpected that AIBP is found in spermatogenesis, considering that the testes operate in a state of immune privilege, and controlling inflammation within the reproductive tract is essential for maintaining an environment conducive to healthy spermatogenesis and overall reproductive function [79].

AIBP expression is often paradoxically upregulated in inflammatory milieu [80]. This may represent a cellular compensatory mechanism to dampen inflammation. In addition, extracellular AIBP levels can be stimulated by APOA1 in 293 cells. Interestingly, AIBP is abundantly expressed in the kidney and secreted AIBP can be found in urine [81]. This implies the potential functional importance of AIBP in maintaining kidney function. On the other hand, AIBP knockout mice do not manifest any obvious defect in kidney function. While it does not suggest an indispensable role of AIBP in the kidney, other compensatory mechanisms may upregulate to mask its effect. On top of this explanation, a careful examination of the kidney ultrastructure and function following a kidney workload with increasing stress may reveal a possible distinction from the wild-type ones.

In addition to the extracellular role of AIBP in anti-inflammation and angiogenic suppression, AIBP also functions intracellularly, particularly in mitochondrial function [82]. While AIBP deletion in mice does not reveal significant impairment of mitochondrial oxidative phosphorylation [83], it attenuates mitophagy and alters the contents of several lipid species [84]. It remains unclear whether the intracellular function of AIBP also contributes to its role in reducing inflammation. This requires employing a non-secreted form of AIBP or transferring wild-type mitochondria into AIBP deficient cells. The intracellular function of AIBP is attributed to the repair of NAD(P)HX, the hydrated form of NAD(P)H, but the causal connection of NAD(P)HX to inflammation remains elusive. This enzymatic activity is hypothesized to be responsible for the neurometabolic disorder found in AIBP mutant patients, the symptoms of this disease often occur following a fever [84,85]. The kinetic measurement of NAD(P)HX content before and after fever in the same patient, coupled with the recapitulation of human neurological disorder in animals following the delivery of the toxic product of NAD(P)HX, will establish a compelling causal relationship between this enzymatic function, NAD(P)HX, and neurometabolic defects. The supplementation of NAD+ or niacin has been suggested to improve disease outcomes [86,87]. Niacin, also known as vitamin B3, plays a significant role in lipid metabolism by effectively increasing HDL-C and lowering LDL-C levels [88]. These effects mimic the role of AIBP in enhancing HDL function [25]. However, a definitive approach to address this disease and to confirm its connection with NAD(P)HX repair is to implement AAV or lipid nanoparticle (LNP)-mediated AIBP overexpression. Because AIBP is a secreted protein, restoring its expression in only a subset of the mutant cells might suffice, as chimeric expression among the cell population coupled with secretion and uptake may restore the AIBP function in mutant cells. In AIBP mutant patients, mortality often follows brain and spinal cord edema, potentially causing irreversible damage to neurons or their underlying circuits. Thus, preserving brain neurons from irreversible injury in those patients is crucial.

Intracellular AIBP has been shown to mitigate atherosclerosis by regulating mitochondrial function and autophagy in macrophages (Figure 1D). AIBP is highly expressed in atherosclerotic lesions, particularly within the mitochondria of macrophages, where it governs PINK1-dependent mitophagy and M1/M2 polarization. AIBP deficiency in bone marrow leads to metabolic disorders, increased macrophage infiltration, inflammation, and accelerated atherosclerosis progression in *Ldlr*^−/−^ mice [89]. At the molecular level, AIBP promotes mitophagy by enhancing the ubiquitination of MFN1/2 through its interaction with PARK2, thereby maintaining mitochondrial quality. AIBP deficiency impairs this process, resulting in mitochondrial dysfunction, the increased generation of reactive oxygen species, reduced autophagy, and enhanced inflammation, ultimately exacerbating lesion development [82]. These findings suggest that AIBP serves as a key regulator of macrophage autophagy and mitochondrial homeostasis. Further studies are needed to determine whether atherogenic stresses influence AIBP/PARK/MFN complex formation and whether extracellular AIBP can be internalized to compensate for mitochondrial AIBP function.

Hematopoietic stem and progenitor cell (HSPC) hyper-proliferation and the resultant monocytosis exacerbate atherosclerosis. During development, AIBP promotes the fate of HSPCs via a SREBP2-dependent mechanism [90,91]. Several studies report the atheroprotective effects of SREBP2 inhibition [92]. In line with this finding, AIBP deficiency increases the atherogenic burden. However, proatherogenic observations of AIBP deficiency appear to be diet dependent. Increasing AIBP levels in hyperlipidemic mice reduces atherosclerosis. Thus, the role of AIBP in the fate of HSPCs in the development of disease can be context dependent. Regardless, AAV-mediated AIBP overexpression in the liver successfully reduces the atherosclerotic burden [93,94]. Both liver and plasma total cholesterol and total triglyceride levels are reduced. The AIBP gain-of-function mutation reduces VLDL without changing the abundance of other lipoprotein fractions [93], which can be partially explained by the increased reverse cholesterol transport [94]. However, whether AIBP also impacts VLDL secretion remains unknown.

AIBP has low immunogenicity in mice [82], allowing for repetitive treatments without eliciting a significant immune response. This characteristic could be advantageous for the development of therapies that require multiple administrations; although its immunogenic profile in humans remains to be thoroughly evaluated.

While AAV-mediated AIBP overexpression has shown promise in mice [93], its application in humans requires further testing and optimization to ensure both efficacy and safety. Additionally, the pharmacokinetics of AIBP in plasma are still unknown, which is essential for determining the correct treatment duration and dosing regimens. Furthermore, because AIBP suppresses neovascularization [74,79], it may not be suitable for patients who require vascular regeneration therapy, highlighting the need for careful patient selection in future clinical applications.

## 7. Targeted Therapies against Specific Inflammatory Signaling: Implications of AIBP-Based Precision Therapy

Mounting evidence suggests that alleviating endothelial inflammation reduces the atherosclerotic burden [95,96]. Inflamed endothelium upregulates cell adhesion molecules, such as E-selectin, ICAM-1 and VCAM-1, which are localized to membrane lipid rafts [97,98,99,100,101]. Notably, many proinflammatory receptors are also localized in lipid rafts, where they facilitate or initiate inflammatory signaling. Conceivably, there is great promise in developing therapeutic components for atherosclerosis that can specifically engage the leukocyte adhesion molecules associated with endothelial inflammation and simultaneously maneuver the cholesterol efflux program to reduce inflammation. 

As discussed above, lipid rafts are essential for a plethora of inflammatory receptor signaling and the disruption of these lipid rafts is predicted to reduce signaling potency. An engineered cholesterol efflux platform that specifically targets particular inflammatory receptors could reduce signaling. This can be achieved by conjugating a targeting peptide or receptor-directed antibody to APOA1 or HDL, as has been reported [102]. Importantly, this platform can be further modified to target a combination of receptors or cell types, such as M1 or M2-type macrophages. Compared to systemic and non-specific targeting, this strategy can be limited to a specific cell type and receptor, thereby minimizing adverse side effects. Targeted therapy may also reduce the effective doses required to achieve therapeutic benefits, increasing the precision and efficacy of the treatment.

Vascular quiescence, characterized by the absence of EC proliferation/migration and vascular leakage, is critical for organ hemostasis and function. Instead of passive acquiescence, active cellular programs dictate vascular quiescence. The accumulating studies have identified numerous pathways controlling vascular quiescence [103,104,105,106,107,108]. Enhancing vascular quiescence could be a potential therapeutic approach to reduce atherosclerosis and CVD.

In addition to ECs, other cells within the vascular wall also contribute to the atherosclerotic burden. While macrophages dominate the inflammatory cell population within atherosclerotic plaque, epigenetic studies, in combination with single-cell RNA-seq analysis, reveal that vascular smooth muscle cells, but not monocytes, contribute to approximately a third of the CD68^+^ macrophages within advanced human plaque [109]. Suppression of this phenotypical transformation of SMCs into macrophage-like cells represents a key approach to reducing vascular inflammation.

## 8. Mitigating Thrombosis

While the rupture of vulnerable plaque still accounts for ~60% of heart attacks, there has been a shift in the clinical manifestations of atherosclerotic plaque following widespread statin therapies. In the post-statin era, patients with ASCVD experience a greater incidence of thrombosis due to de-endothelization [110]. Thrombosis, and the subsequent embolism, pose significant risk of MI, stroke, or peripheral arterial disease. Endothelial inflammation in the atherosclerotic plaque contributes to thrombosis [111,112,113]. In addition, hyperlipidemia increases thrombogenic potential by (1) promoting platelet biogenesis and activation, rendering them procoagulant [114,115], and (2) precipitating HSPC-derived monocytosis [116,117,118]. Platelets can facilitate the recruitment of the resulting monocytes to atherosclerotic lesions [119], enhancing acute thrombosis [120,121]. The Jain laboratory at Texas A&M has pioneered the study of thrombosis using the vessel-chip model [122], with the ability of engineering, the chip mimics the differential flow dynamics of the human coronary artery, such as pro-atherogenic disturbed flow and atheroprotective laminar flow. Furthermore, the Jain laboratory has established a method to generate patient-derived blood outgrowth ECs. Applying blood-derived ECs from individual CVD patients to vessel-chip engineering will facilitate the development of personalized, patient-oriented therapies [123].

## 9. Conclusions

Reducing endothelial inflammation is predicted to enhance vascular integrity, reduce vascular inflammation, and consequently lower the atherosclerotic burden. Emerging evidence underscores that rational design and engineering of targeted cholesterol efflux mechanisms may yield improved outcomes for patients with ASCVD. The advent of bioengineering and synthetic biology will facilitate the design of such therapeutic ends with broad impacts.

Moreover, our proposed cell type-specific targeting approach holds promise not only for ASCVD, but also for the treatment of diseases affecting other organs. This strategy aims to deliver safer and more efficacious doses, thereby minimizing adverse effects and enhancing therapeutic benefits. As such treatments advance, integrating these innovative methodologies will be crucial in addressing the complex challenges posed by atherosclerosis and related vascular conditions.

## Figures and Tables

**Figure 1 ijms-25-10314-f001:**
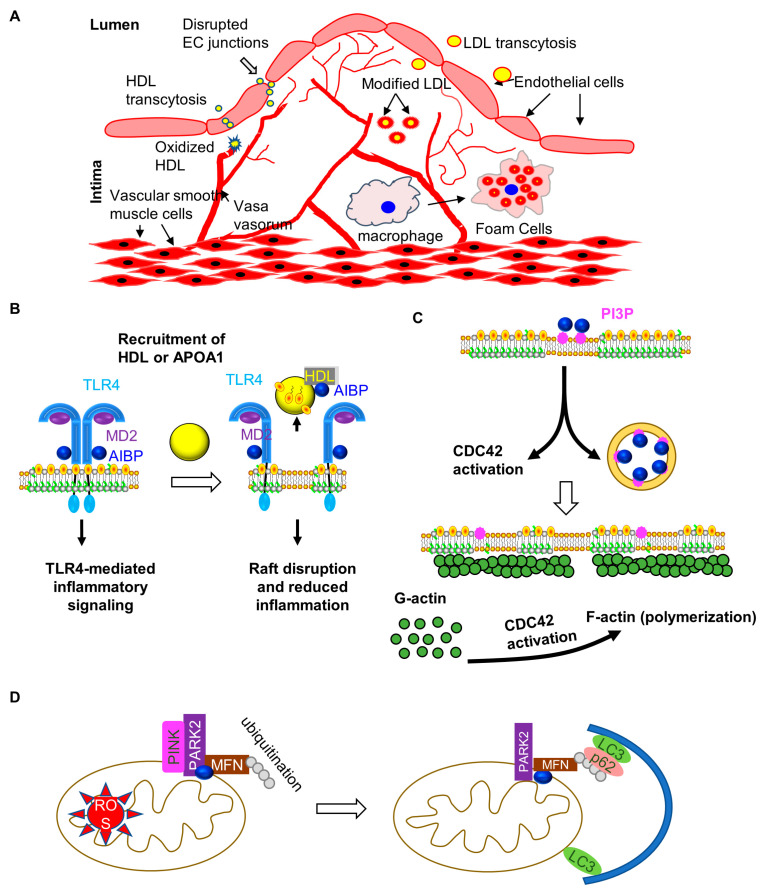
(**A**). Lipoprotein penetration and retention in the sub-endothelium. Hyperlipidemia contributes to greater LDL and HDL presence in the subendothelial space, where they are rendered proinflammatory. These modified lipoproteins will be taken up by macrophages, which subsequently become lipid-laden foam cells. (**B**). AIBP impedes TLR4-mediated inflammatory signaling in myeloid cells. TLR4 dimerization is essential for the innate inflammatory response. AIBP binds TLR4, enabling a targeted cholesterol efflux, which disrupts TLR4 dimerization and represses signaling. This AIBP-mediated effects result in inhibited inflammatory responses. (**C**). AIBP regulation of lipid rafts via cytoskeleton rearrangement. AIBP binds PI(3)P, activating CDC42, and triggering actin polymerization, which consequently reduces the lipid raft levels. (**D**). AIBP regulates mitophagy. Mitochondrial AIBP binds PARKIN and MFN and increases the ubiquitination of MFN. Ubiquitinated MFN will recruit P62, which in turn facilitates the LC3 interactions, thereby targeting these mitochondria for degradation via autophagy.

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
