# Peer review of "Mitigating Vascular Inflammation by Mimicking AIBP Mechanisms: A New Therapeutic End for Atherosclerotic Cardiovascular Disease"

_ijms, 2024, doi:10.3390/ijms251910314_

Round 1

Reviewer 1 Report

Comments and Suggestions for Authors

1. The review addresses a timely and relevant topic in atherosclerosis management by focusing on endothelial inflammation. The introduction of APOA1 binding protein (AIBP) as a novel therapeutic strategy is intriguing and provides a fresh perspective. 

2. The focus on endothelial inflammation as a key driver of atherosclerosis is relevant, as current therapies do not specifically target endothelial cell (EC) inflammation. Clearly articulate the clinical relevance of targeting endothelial inflammation specifically, which would provide a stronger rationale for the review.

3. Discuss the possible challenges and considerations for translating AIBP research into clinical practice.

4. Address potential concerns such as off-target effects, immune responses, or issues with delivery mechanisms for AIBP in clinical settings.

Comments on the Quality of English Language

Minor editing of English language required.

Author Response

  1. The review addresses a timely and relevant topic in atherosclerosis management by focusing on endothelial inflammation. The introduction of APOA1 binding protein (AIBP) as a novel therapeutic strategy is intriguing and provides a fresh perspective. 

Response: We appreciate these complimentary remarks.

  1. The focus on endothelial inflammation as a key driver of atherosclerosis is relevant, as current therapies do not specifically target endothelial cell (EC) inflammation. Clearly articulate the clinical relevance of targeting endothelial inflammation specifically, which would provide a stronger rationale for the review.

Response: We have added a stronger rationale in the summary as following: “Targeting endothelial inflammation is clinically significant because it can mitigate the root cause of atherosclerosis, potentially preventing disease progression while reducing the side effects associated with broader anti-inflammatory treatments.”

  1. Discuss the possible challenges and considerations for translating AIBP research into clinical practice. Address potential concerns such as off-target effects, immune responses, or issues with delivery mechanisms for AIBP in clinical settings.

Response: We have added the perspective and challenges of translating AIBP research (line 373-383 in the revised manuscript).

  1. Comments on the Quality of English Language

Minor editing of English language required.

Response: We have checked the text carefully and revised the text.

Reviewer 2 Report

Comments and Suggestions for Authors

Dear Authors,

You raised on of the currently most crucial topics in vascular medicine, which is the role of inflammatory components and its inhibition. In general, that paper is well written, considering multiple mechanisms of vascular inflammation. I am really glad that you mentioned colchicine, as in the last couple of years, this was the subject of multiple discussions on its effectiveness. There is recent review on this subject with all of advantages and drawbacks of colchicine in cardiovascular disease, I would suggest reading it and include in your paper (10.1136/heartjnl-2023-323177).

Unfortunately, you totally omitted the eicosanoid pathway. There is quite recent systematic review on the leukotrienes and their impact on cardiovascular disease with analysis of multiple atherosclerotic locations (10.5603/AA.2022.0013). The leukotrienes, especially LTE 4 was proved to not only be related to the occurrence of restenosis after endovascular treatment in peripheral arterial disease, but also predict such events, indicating that the LTE4 elevation is not the result of restenosis but its cause. (10.1016/j.atherosclerosis.2016.04.013). Another study showed that significance of changes in life quality after revascularization is strictly related to vascular inflammation, especially on the eicosanoid pathway (10.3390/jcm12103412). So far, there was no trail regarding inhibition of leukotrienes in atherosclerosis, until now, in which montelukast is used (https://clinicaltrials.gov/study/NCT04277702). Considering all this information I would suggest expanding the paper by additional paragraph related to eicosanoid-based inflammatory pathway in atherosclerosis. 

Also, I would suggest some minor language corrections. 

Kind regards

Comments on the Quality of English Language

Minor language corrections suggested. 

Author Response

  1. You raised on of the currently most crucial topics in vascular medicine, which is the role of inflammatory components and its inhibition. In general, that paper is well written, considering multiple mechanisms of vascular inflammation. I am really glad that you mentioned colchicine, as in the last couple of years, this was the subject of multiple discussions on its effectiveness. There is recent review on this subject with all of advantages and drawbacks of colchicine in cardiovascular disease, I would suggest reading it and include in your paper (10.1136/heartjnl-2023-323177).

Response: We thank you for complimentary remarks and great suggestions.

We have added the drawbacks of colchicine application in CVD as following (lines 215-221).

  1. Unfortunately, you totally omitted the eicosanoid pathway. There is quite recent systematic review on the leukotrienes and their impact on cardiovascular disease with analysis of multiple atherosclerotic locations (10.5603/AA.2022.0013). The leukotrienes, especially LTE 4 was proved to not only be related to the occurrence of restenosis after endovascular treatment in peripheral arterial disease, but also predict such events, indicating that the LTE4 elevation is not the result of restenosis but its cause. (10.1016/j.atherosclerosis.2016.04.013). Another study showed that significance of changes in life quality after revascularization is strictly related to vascular inflammation, especially on the eicosanoid pathway (10.3390/jcm12103412). So far, there was no trail regarding inhibition of leukotrienes in atherosclerosis, until now, in which montelukast is used (https://clinicaltrials.gov/study/NCT04277702). Considering all this information I would suggest expanding the paper by additional paragraph related to eicosanoid-based inflammatory pathway in atherosclerosis. 

Response: We have added a section of eicosanoid (lines 280-300).

  1. Also, I would suggest some minor language corrections. 

Response: Please see response in # 4.

Reviewer 3 Report

Comments and Suggestions for Authors

This manuscript reviews current therapies designed to reduced cardiovascular outcomes as well as emerging therapies targeting AIBP. The review first describes several mechanisms involved in atherosclerosis development, such as vascular integrity, shear stress, etc. However, little information is provided for each mechanism. A more detailed description for each process would be more informative. For example, the role of H3K27me3 methylation is unclear, as described. In the following sections, authors described current medications reducing systemic inflammation, like statins. The major point of the review seems to describe the role of AIBP in reducing vascular inflammation. However, AIBPO is not in the title of the manuscript.

There are other reviews also dealing with the role of AIBP in inflammation and CVD that are comparable to this manuscript. Overall, this manuscript offers little novelty or new information compared with other published reviews by the same authors and others. For example, PMIDS: 33811580, 33230630, 26634023.

Models could be used to describe the extracellular and intracellular functions of AIBP.

There are also several issues with citations and organization:  

·      Define RAC1 and ALK1 in line 58

·      Revise SR-BI vs SR-B1 for consistency

·      Revise “ALK1 is identified” in line 58. It should be “was”

·      Abbreviations are missing in different sections of the manuscript. For example, YAP/TAZ, NLRP3, EZH2, JMJD3 were not defined in section 2.3

·      The sentence in lines 171-172 is a repetition of the information in the previous sentence.

·      In line 259, revise the statement for mTOR on autophagy. Inhibition, not activation of mTOR promotes autophagy/mitophagy.

·      “yet” can be removed from the end of the sentence in line 277.

·      What PIP1 stands for in line 281?

·      In line 282 revise “constituent”

·      The statement in lines 28-283 is not accurate. Reference # 69 investigated the role of AIBP in lipid rafts not endocytosis.

·      It is unclear what mechanism is the author referring to in line 285. If the mechanism is endocytosis, then reference.

·      Citations should be carefully revised as many statements lack proper citations. For example, the information in lines 300-303 in which the role of AIBP in the kidney is discussed.

·      Revise reference #66.

Comments on the Quality of English Language

There are some issues with tense usage (present/past) throughout the manuscript 

Author Response

  1. This manuscript reviews current therapies designed to reduced cardiovascular outcomes as well as emerging therapies targeting AIBP. The review first describes several mechanisms involved in atherosclerosis development, such as vascular integrity, shear stress, etc. However, little information is provided for each mechanism. A more detailed description for each process would be more informative. For example, the role of H3K27me3 methylation is unclear, as described.

Response:  In the revised manuscript, we have updated some mechanisms. We also address the role of H3K27me3 methylation as following: “As a result, it reduces JMJD3 expression and attenuates proinflammatory JMJD3 signaling. (lines 91-92)”

  1. In the following sections, authors described current medications reducing systemic inflammation, like statins. The major point of the review seems to describe the role of AIBP in reducing vascular inflammation. However, AIBP is not in the title of the manuscript.

Response:  We have now added AIBP to the title.

There are other reviews also dealing with the role of AIBP in inflammation and CVD that are comparable to this manuscript. Overall, this manuscript offers little novelty or new information compared with other published reviews by the same authors and others. For example, PMIDS: 33811580, 33230630, 26634023. Models could be used to describe the extracellular and intracellular functions of AIBP.

 Response:  Thank you for pointing out the reviews we wrote years ago. We think it is the time to provide an update for AIBP research.

There are also several issues with citations and organization:  

  1. Define RAC1 and ALK1 in line 58

Response:  We have defined RAC and ALK1 in lines 62-65.

  1. Revise SR-BI vs SR-B1 for consistency

Response:  We have changed all to SR-BI.

  1. Revise “ALK1 is identified” in line 58. It should be “was”.

Response:  We have changed it to “was”.

  1. Abbreviations are missing in different sections of the manuscript. For example, YAP/TAZ, NLRP3, EZH2, JMJD3 were not defined in section 2.3

Response:  We have defined all these in the revised manuscript.

  1. The sentence in lines 171-172 is a repetition of the information in the previous sentence.

Response:  We have removed the repetitive information.

  1. In line 259, revise the statement for mTOR on autophagy. Inhibition, not activation of mTOR promotes autophagy/mitophagy.

Response:  We apologize for this. We have now changed to “inhibiting” (line 275).

  1. “yet” can be removed from the end of the sentence in line 277.

Response:  We have removed “yet”.

  1. What PIP1 stands for in line 281?

Response:  We apologize for this. We meant phosphatidylinositol 3-phosphate (PI(3)P). It was fixed in the revised manuscript.

  1. In line 282 revise “constituent”

Response:  We apologize for this. It was changed to “consistent”.

  1. The statement in lines 28-283 is not accurate. Reference # 69 investigated the role of AIBP in lipid rafts not endocytosis.

Response:  We apologize for the inaccuracy. We have now placed Reference #69 in the correct context, reflecting its focus on the role of AIBP in lipid rafts rather than endocytosis.

  1. It is unclear what mechanism is the author referring to in line 285. If the mechanism is endocytosis, then reference.

Response:  We apologize for this. We have added the correct reference (#73). The reference is:

Takahashi, Y.; Smith, J.D. Cholesterol efflux to apolipoprotein AI involves endocytosis and resecretion in a calcium-dependent pathway. Proc Natl Acad Sci U S A 1999, 96, 11358-11363, doi:10.1073/pnas.96.20.11358.

  1. Citations should be carefully revised as many statements lack proper citations. For example, the information in lines 300-303 in which the role of AIBP in the kidney is discussed.

Response:  Now the reference is added (#78).

Ritter, M.; Buechler, C.; Boettcher, A.; Barlage, S.; Schmitz-Madry, A.; Orso, E.; Bared, S.M.; Schmiedeknecht, G.; Baehr, C.H.; Fricker, G.; et al. Cloning and characterization of a novel apolipoprotein A-I binding protein, AI-BP, secreted by cells of the kidney proximal tubules in response to HDL or ApoA-I. Genomics 2002, 79, 693-702, doi:10.1006/geno.2002.6761.

  1. Revise reference #66.

Response:  Now the correct reference is added.

Round 2

Reviewer 2 Report

Comments and Suggestions for Authors

Dear Authors,

Thank you for the corrections. At this point, I do not have any further requests or questions.

Kind regards

Author Response

We appreciate the time and insightful comments from this reviewer.

Reviewer 3 Report

Comments and Suggestions for Authors

Although the authors responded to most previous critics, some issues remain. For example, from a previous comment about endocytosis required for the function of AIBP, the sentence in lines 345-346 is an overstatement of the findings in reference #72. The paper shows that AIBP is internalized, but it did not demonstrate that the internalization of this molecule was required to disrupt the lipid rafts. As suggested in the previous review, a model is needed to summarize the several mechanisms by which AIBP can protect the cardiovascular system (intracellular and extracellular pathways). However, a model was not provided in the revised manuscript. Also, the invalid citation in line 733 was not corrected.

Since this group has published a few reviews on this topic, it is important to mention in the introduction the new developments in this area that will be discussed in the review.  This reviewer also had concerns about missing references and grammatical mistakes. There are still many issues that need correction. For example:

·      Remove citations from the abstract.

·      Define SGLT2 in the abstract.

·      Remove the s from reduces in line 21.

·      Define SR-BI in line 76.

·      Revise (50% mol% cholesterol) in line 124

·      Define SREBP2 in line 152

·      A citation is needed for the statement in lines 154-156. It is unclear if reference #31 also applies to this statement.

·      Two meta-analyses are mentioned in line 164 with only 1 reference.

·      A sentence is required in line 167 before the i, ii, and iii paragraphs. Something that relates to the mechanisms of action.

·      The section in lines 194-199 needs references for each sentence. Each of these mechanisms needs the citation in which the effects were reported.

·      Define NPC1L1 in line 210

·      References are needed for lines 210-214, 270-277, 306-312, 318-320, 328-340, 360-362, 373-374, 385-386, 402-403.

·      Revise medications in line 239

·      Remove the “s” from regulates in line 307 and from exacerbates in line 465.

·      Unclear what “this study” in lines 312-313 refers to.

·      Define VEGFR2 and NOTCH in lines 456-457

·      Replace “poses” with “pose” in line 494.

Many other mistakes are not mentioned in this list and require a line-by-line revision from the authors.

Comments on the Quality of English Language

same mistakes are listed in the commnets to authors

Author Response

We have updated the latest knowledge on AIBP and generated a new Figure 1.

Additionally, we have made the corresponding changes requested by the reviewer.

We also updated several references. Since this journal prefers citations from publications within the past 5 years, we occasionally cited recently published review papers.

We are grateful for the reviewer’s insightful comments, which have greatly improved this manuscript.